

# Genome-wide identification and characterization of *NHL* gene family in response to alkaline stress, ABA and MEJA treatments in wild soybean (*Glycine soja*)

Xu Zhang[1,*], Yongguo Xue[2,*], Haihang Wang[1], Zaib_un Nisa[3], Xiaoxia Jin[1], Lijie Yu[1], Xinlei Liu[2], Yang Yu[4] and Chao Chen[1]

[1] Harbin Normal University, Harbin, Heilongjiang, China
[2] Heilongjiang Provincial Academy of Agricultural Sciences, Harbin, Heilongjiang, China
[3] University of Lahore, Lahore, Pakistan
[4] Shenyang University, Shenyang, China
[*] These authors contributed equally to this work.

Corresponding authors
Yang Yu, yuyang_syu@163.com
Chao Chen, chchao@hrbnu.edu.cn

## ABSTRACT

**Background.** *NDR1/HIN1-like* (*NHL*) family genes are known to be involved in pathogen induced plant responses to biotic stress. Even though the *NHL* family genes have been identified and characterized in plant defense responses in some plants, the roles of these genes associated with the plant abiotic stress tolerance in wild soybean is not fully established yet, especially in response to alkaline stress.

**Methods.** We identified the potential *NHL* family genes by using the Hidden Markov model and wild soybean genome. The maximum-likelihood phylogenetic tree and conserved motifs were generated by using the MEME online server and MEGA 7.0 software, respectively. Furthermore, the syntenic analysis was generated with Circos-0.69. Then we used the PlantCARE online software to predict and analyze the regulatory *cis*-acting elements in promoter regions. Hierarchical clustering trees was generated using TM4: MeV4.9 software. Additionally, the expression levels of *NHL* family genes under alkaline stress, ABA and MEJA treatment were identified by qRT-PCR.

**Results.** In this study, we identified 59 potential *NHL* family genes in wild soybean. We identified that wild soybean *NHL* family genes could be mainly classified into five groups as well as exist with conserved motifs. Syntenic analysis of *NHL* family genes revealed genes location on 18 chromosomes and presence of 65 pairs of duplication genes. Moreover, *NHL* family genes consisted of a variety of putative hormone-related and abiotic stress responsive elements, where numbers of methyl jasmonate (MeJA) and abscisic acid (ABA) responsive elements were significantly larger than other elements. We confirmed the regulatory roles of *NHL* family genes in response to alkaline stress, ABA and MEJA treatment. In conclusion, we identified and provided valuable information on the wild soybean *NHL* family genes, and established a foundation to further explore the potential roles of *NHL* family genes in crosstalk with MeJA or ABA signal transduction mechanisms under alkaline stress.

## INTRODUCTION

NHL (*NDR1/HIN1-like*) family genes are previously reported to be involved in plant defense response against pathogens, such as *Phytophthora infestans*, *Botrytis cinerea*, *Pseudomonas syringae* (*Chen et al., 2018*; *Chong et al., 2008*; *Varet et al., 2002*). Previous studies revealed that there are at least 29 *NHL* family members that show homology to *NDR1* and *HIN1* gene in Arabidopsis based on the non-redundant GenBank database and then increased this number to 45 upon the completed genome sequencing of Arabidopsis, suggesting a potential role for Arabidopsis NDR1/HIN1-like family genes in plant-pathogen interactions. The 45 NHL family genes were divided into four groups and shared three conserved sequence motifs. (*Dormann et al., 2000*; *Zheng et al., 2004*). Analysis of amino acid sequence reveals that a lot of NHL family proteins contain a specific conserved late embryogenesis abundant (LEA) domain and putative transmembrane domain (*Dormann et al., 2000*; *Liu et al., 2020b*).

During the last three decades, many NHL family members were isolated and identified to play important roles in triggering plants defense resistance. The *HIN1* gene (harpin-induced gene), which could be rapidly activated and elicit HR (hypersensitive response) phenomenon when plants exposed to bacterial pathogens (*e.g.*, *Pseudomonas syringae* pv. syringae), was first isolated in tobacco (*Gopalan et al., 1996*). The *NDR1* gene (non-race-specific disease resistance gene), which shows sequence similarity with tobacco *HIN1*, was first identified to play distinct roles in response to both bacterial and fungal pathogen resistance in Arabidopsis (*Century, Holub & Staskawicz, 1995*; *Takahashi et al., 2004*). Overexpression of the NHL3 gene could enhanced the plant resistance to *Pseudomonas syringae* pv. *tomato* DC3000, which was a membrane-localized protein in Arabidopsis (*Varet et al., 2003*). In soybean (*Glycine max*), two homologs of *Arabidopsis NDR1* gene named *GmNDR1a* and *GmNDR1b* were identified (*Selote et al., 2014*). This study showed that the NDR1 protein could interact with RIN4 to play roles in resistance to *Pseudomonas syringae*. The function role of *GmNDR1b*, also named *Gm-NDR1-1*, was further determined that played important roles in impairing root pathogenic nematode *Heterodera glycines* and *Meloidogyne incognita* (*McNeece et al., 2017*). The *StPOTHR1*, a NHL family member in potato, could enhanced plants resistance to Phytophthora infestans through effecting the MAP kinase signaling process by interacting with NbMKK5L (*Chen et al., 2018*). Overexpression of the pepper *CaNHL4* enhanced the expression of salicylic acid (SA)-related and jasmonic acid (JA)-related genes, increased ROS production, and inhibited the infection of the pathogens (*Liu et al., 2020a*). The interaction of ToxA with NHL10 protein could induce cell death under plant pathogen stress in wheat (*Dagvadorj et al., 2022*)

Saline-alkaline soils are known to have high content of sodium, bicarbonates and high pH, which consequently causes growth retardation and ultimately leads to death of plants growing in such soils. The total of 434 million ha of global land is affected by alkaline soils (*Jin et al., 2006*; *Wang et al., 2008*). In comparison with neutral salts stress, alkaline stress exerts more harmful effects on plant growth (*Yang et al., 2008*). Alkaline stress can inhibit photosynthesis, N and sugar metabolism, as well as limits the absorption of ions, such as $H_2PO_4^-$, $Cl^-$, $Al^{3+}$ and $Fe^{2+}$ (*Vondrackova et al., 2015*).

It has been documented that *NHL* family genes also play distinct roles in plant abiotic stress resistance. In pepper (*Capsicum annuum* L.), fifteen *NHL* genes were identified in a genome-wide analysis and the responses of these genes were characterized under different abiotic stresses (*Liu et al., 2020b*). A stress-inducing *NHL* member, *BnNHL18A*, was isolated from *Brassica napus*, which displayed roles in response to different treatments including NaCl, $H_2O_2$, as well as ethephon and SA (*Lee et al., 2006*). In Arabidopsis, overexpression of NHL6 increased the sensitivity to salt, osmotic and ABA treatment, and NHL6 could affect the seed germination and early seedling development under these stresses-induced ABA signaling (*Bao et al., 2016*). Furthermore, studies have shown that some saline or alkaline stress induced transcription factors co-expresses with NHL network to regulate the salt or alkali response (*Liu et al., 2020c*). The soybean NHLs have been identified to play important roles in regulating seed germination under chilling stress and with ABA treatment (*Wang et al., 2022*). However, little is known about the wild soybean NHL family genes in response to environment stresses, especially under alkaline stress.

Soybean has been adopted as important crops in the world, particularly for protein and oil production (*Yu et al., 2018*). Wild soybean, as the ancestor of cultivated soybean, showed a better adaptation to various abiotic stress, such as salt, drought and alkaline. Therefore, the wild soybean has been suggested as a valuable sources to improve the agronomic traits of soybean (*Wen et al., 2009*). In previous studies, we have identified a highly adaptable saline-alkali soil tolerant wild soybean (*Glycine soja*) line (G07256). It can survive well in the saline-alkali soil (*Ge et al., 2010*). By using transcriptome data, we identified some candidate genes in response to alkaline stress. In this study, the NHL family genes in wild soyabean genome were identified and their expression was investigated under the influence of growth hormones in alkaline stress which may enhance the stress responses.

## MATERIAL AND METHODS

### Identification of NHL family genes in wild soybean genome

To identify all potential genes encoding NHL family genes in wild soybean genome, a Hidden Markov model was first established by using the Arabidopsis and soybean amino acid sequences of NHL family genes as queries (*Gopavajhula et al., 2013*; *Wang et al., 2022*). The HMM profile (build 2.3.2) was further used to search in wild soybean genome database to get similar sequences (*Finn, Clements & Eddy, 2011*). Then, the potential genes were identified after removing the overlapping genes and incomplete domains genes through Pfam and SMART database (*Finn et al., 2016*). ExPASy (https://web.expasy.org/compute_pi/) was used to predict the molecular weight and isoelectric point values of NHL family proteins (*Artimo et al., 2012*).

### Bioinformatics analysis of NHL family genes

The conserved motifs of all the potential *NHL* family genes were identified by MEME online server (http://meme-suite.org/) (*Bailey et al., 2009*). The maximum-likelihood phylogenetic tree was constructed by software MEGA 7.0 (*Kumar et al., 2008*) software. Then, the TBtools software was used to combine the conserved motifs and phylogenetic tree. The syntenic analysis was generated with Circos−0.69 (http://circos.ca/) (*Krzywinski et*
al., 2009). To analyze the potential regulatory *cis*-acting elements in the promoters of *NHL* genes, 3000 bp upstream sequences of the above-mentioned genes were extracted based on the genome database (*Sun et al., 2014*). Then we used the PlantCARE online software (http://bioinformatics.psb.ugent.be/webtools/plantcare/html/) to predict and analyze the regulatory cis-acting elements in promoter regions (*Lescot et al., 2002*). To examine the expression profiles of *NHL* family genes under alkaline stress, wild soybean transcriptome data was downloaded (*DuanMu et al., 2015*) and hierarchical clustering trees were generated using TM4: MeV4.9 software (*Saeed et al., 2006*).

## Plant material, growth condition and stress treatment

The wild soybean cultivar DN50 was grown in 1/4 Hoagland nutrient solutions in the growth chamber with 22−28 °C room temperature, 70–80% relative humidity and 8 h dark/16 h light. The healthy and plump seeds were rinsed with 75% ethanol for 1 min, and then washed with sterile water before germination (*Qiao et al., 2020*). After two days, the germinated seedlings were transferred in to 1/4 strength Hoagland nutrient solutions to be cultured and the nutrient solutions were changed every two days. Twelve days later, the young seedlings were treated with alkaline (NaHCO$_3$) stress or exogenous hormones (ABA and MeJA), respectively. For alkaline stress, the 12-days old seedlings were transferred into 1/4 Hoagland solution with 50 mM NaHCO$_3$. For exogenous hormones treatment, the 12-days old seedlings were transferred into 1/4 Hoagland solution with 50 μM ABA or 1/4 Hoagland solution with 50 μM MeJA. The roots were harvested and stored in liquid nitrogen at 0 h, 1 h and 3 h after treatment for RNA extraction.

## Transcript expression analysis by qRT-PCR

Total RNA was extracted with the OminiPlant RNA isolation kit (Kangwei), and the cDNAs were synthesized using the First Stand cDNA Synthesis kit (Toyobo) for qRT-PCR. The qRT-PCR was then performed with UtraSYBR Mixture (Baioleibo) and ABI 7500 sequencer. The primers of wild soybean *NHL* genes and *GsGADPH* were listed in Table S1 which are used in our study. Here the *GsGADPH* gene was used as an internal control in wild soybean (*Huis, Hawkins & Neutelings, 2010*). The qRT-PCR data was calculated with three independent biological replicates using $2^{-\Delta\Delta CT}$ method and Student's *t*-test.

# RESULTS

## Identification of NHL genes in wild soybean

To identify the *NHL* family genes in wild soybean, the amino acid sequences of the *NHL* family genes from Arabidopsis and soybean were queried against the wild soybean genome via BLAST from NCBI. A total of 208 NHL candidate sequences were obtained based on the Hidden Markov model. All candidate sequences were then subjected to Pfam and SMART database to remove the redundant sequences or incomplete domain sequences. As a result, 59 genes were obtained as potential *NHL* family genes in wild soybean genome.

As shown in Table 1, 59 predicted wild soybean *NHL* genes were named based on the location within the reference genome from *GsNHL1* to *GsNHL59*. Then the chemical properties of these proteins were determined including the protein sequence lengths,

molecular weights (MW), and theoretical isoelectric points (pI). The protein sequence length was ranged from 149 (*GsNHL33*) to 348 (*GsNHL17*) amino acids residues. The MW varied from 16.37778 (*GsNHL33*) to 40.00232 (*GsNHL17*) kDa and the pI values ranged from 7.82 (*GsNHL31*) to 10.24 (*GsNHL43*).

## Phylogenetic and conserved motifs analysis of wild soybean NHL genes

To confirm the evolutionary relationships of *NHL* family genes in wild soybean, a maximum-likelihood phylogenetic tree was constructed with the full-length protein sequences from soybean, Arabidopsis and wild soybean. The result showed that the genes could be divided into six groups and the other seven ungroup genes (Fig. S1).

Based on conserved motif sequences, the 59 wild soybean *NHL* family genes could be further classified into five groups (group 1a, group 1b, group 2, group 3a and group 3b) and the other 11 ungroup genes (Fig. 1A). MEME motif analysis also revealed that wild soybean *NHL* family proteins shared ten conserved motif sequences (Fig. 1B, Fig. S2). Most of wild soybean *NHL* family proteins contain conserved motif 1 and motif 2. Interestingly, we found that some wild soybean *NHL* family proteins in the same group shared a similar motif composition. For example, motif 5 is mainly present within group 1b, group 2 and group 3b genes, while most genes of group 1a and group 1b contain motif 7 and 9. The motif 3 only located in group 1a genes. The similar motif arrangement among the proteins of wild soybean *NHL* family suggested that the protein architecture was conserved within subgroups, which indicated that the proteins in the same group may have similar function in plant development and resistance responses under stress conditions. However, functions of these conserved motifs are still need to be further explored.

## Chromosomal locations and syntenic analysis

The analysis of gene duplication events could drive the potential evolution mechanisms of the wild soybean *NHL* family genes. In this study, 59 wild soybean *NHL* family genes were found randomly distributed among 18 chromosomes, with the exception of 8 and 17 (Fig. 2). Gene duplication plays significant roles in plant organismal evolution and functional diversity (*Bowers et al., 2003*). Further, a total of 65 pairs of *NHL* syntenic paralogs were identified in wild soybean genome. These results indicated that the wild soybean *NHL* family have been exhibited a high gene family expansion.

## Identification of cis-acting elements of NHL gene promoters in wild soybean

To explore the potential roles of wild soybean *NHL* family genes in response to abiotic stress, the promoter sequences 3 kb upstream regions of the ATG were predicted using information within the PlantCARE online tool. The results showed that the wild soybean *NHL* family genes displayed a variety of putative hormone-related and abiotic stress responsive elements (Fig. 3, Table S2). For example, the plant hormone-related responsive elements include Methyl jasmonate (MeJA), abscisic acid (ABA, ABRE), gibberellin (GA), salicylic acid (SA) and Auxin responsive elements. Interestingly, we found that the numbers of MeJA and ABA responsive elements were significantly larger than the other plant hormone responsive

**Table 1  Protein information of *NHL* family genes in wild soybean.**

| Number | Gene ID | Gene Name | Chr | Amino acid residues | MW (kDa) | pI |
|---|---|---|---|---|---|---|
| 1 | GsNHL1 | GlysoPI483463.01G116400 | 1 | 230 | 25.81956 | 8.95 |
| 2 | GsNHL2 | GlysoPI483463.01G198800 | 1 | 190 | 21.52359 | 10.15 |
| 3 | GsNHL3 | GlysoPI483463.02G162500 | 2 | 208 | 23.91979 | 9.79 |
| 4 | GsNHL4 | GlysoPI483463.02G162700 | 2 | 245 | 27.62919 | 9.26 |
| 5 | GsNHL5 | GlysoPI483463.02G199000 | 2 | 256 | 28.76769 | 10.12 |
| 6 | GsNHL6 | GlysoPI483463.02G230000 | 2 | 274 | 30.45435 | 9.86 |
| 7 | GsNHL7 | GlysoPI483463.03G161800 | 3 | 222 | 24.93808 | 9.58 |
| 8 | GsNHL8 | GlysoPI483463.03G162100 | 3 | 208 | 24.13909 | 9.46 |
| 9 | GsNHL9 | GlysoPI483463.03G162300 | 3 | 230 | 26.77082 | 9.14 |
| 10 | GsNHL10 | GlysoPI483463.03G162400 | 3 | 204 | 23.49148 | 9.53 |
| 11 | GsNHL11 | GlysoPI483463.03G162500 | 3 | 210 | 23.77761 | 9.81 |
| 12 | GsNHL12 | GlysoPI483463.03G213100 | 3 | 239 | 26.31572 | 9.37 |
| 13 | GsNHL13 | GlysoPI483463.03G218900 | 3 | 198 | 21.55946 | 9.83 |
| 14 | GsNHL14 | GlysoPI483463.03G162200 | 3 | 228 | 26.24044 | 9.50 |
| 15 | GsNHL15 | GlysoPI483463.04G093700 | 4 | 256 | 28.40712 | 9.51 |
| 16 | GsNHL16 | GlysoPI483463.04G185500 | 4 | 211 | 23.22106 | 9.69 |
| 17 | GsNHL17 | GlysoPI483463.05G146000 | 5 | 348 | 40.00232 | 10.00 |
| 18 | GsNHL18 | GlysoPI483463.05G185300 | 5 | 214 | 22.94057 | 9.54 |
| 19 | GsNHL19 | GlysoPI483463.06G095800 | 6 | 260 | 28.63512 | 9.04 |
| 20 | GsNHL20 | GlysoPI483463.06G124800 | 6 | 224 | 25.31161 | 8.97 |
| 21 | GsNHL21 | GlysoPI483463.07G008800 | 7 | 255 | 27.95721 | 9.68 |
| 22 | GsNHL22 | GlysoPI483463.07G045200 | 7 | 253 | 28.11260 | 9.46 |
| 23 | GsNHL23 | GlysoPI483463.07G092400 | 7 | 210 | 24.10902 | 9.43 |
| 24 | GsNHL24 | GlysoPI483463.07G092500 | 7 | 208 | 24.15102 | 9.82 |
| 25 | GsNHL25 | GlysoPI483463.09G134000 | 9 | 204 | 23.68964 | 9.53 |
| 26 | GsNHL26 | GlysoPI483463.09G151000 | 9 | 315 | 34.59544 | 9.85 |
| 27 | GsNHL27 | GlysoPI483463.10G070800 | 10 | 247 | 27.67418 | 9.09 |
| 28 | GsNHL28 | GlysoPI483463.10G070900 | 10 | 216 | 24.78271 | 8.71 |
| 29 | GsNHL29 | GlysoPI483463.10G071000 | 10 | 228 | 26.25031 | 9.64 |
| 30 | GsNHL30 | GlysoPI483463.10G071100 | 10 | 210 | 24.00494 | 10.02 |
| 31 | GsNHL31 | GlysoPI483463.10G071200 | 10 | 200 | 22.61333 | 7.82 |
| 32 | GsNHL32 | GlysoPI483463.10G071700 | 10 | 223 | 24.65133 | 9.10 |
| 33 | GsNHL33 | GlysoPI483463.10G105800 | 10 | 149 | 16.37778 | 9.39 |
| 34 | GsNHL34 | GlysoPI483463.10G214600 | 10 | 228 | 26.04372 | 9.56 |
| 35 | GsNHL35 | GlysoPI483463.11G015000 | 11 | 179 | 19.59170 | 9.55 |
| 36 | GsNHL36 | GlysoPI483463.11G145400 | 11 | 215 | 23.56234 | 9.96 |
| 37 | GsNHL37 | GlysoPI483463.11G186300 | 11 | 246 | 27.22966 | 8.88 |
| 38 | GsNHL38 | GlysoPI483463.12G084400 | 12 | 214 | 23.57643 | 9.96 |
| 39 | GsNHL39 | GlysoPI483463.12G153200 | 12 | 222 | 24.646.62 | 9.34 |

**Table 1** (*continued*)

| Number | Gene ID | Gene Name | Chr | Amino acid residues | MW (kDa) | pI |
|---|---|---|---|---|---|---|
| 40 | *GsNHL40* | *GlysoPI483463.12G178100* | 12 | 218 | 24.24817 | 9.81 |
| 41 | *GsNHL41* | *GlysoPI483463.13G250400* | 13 | 233 | 26.33392 | 7.93 |
| 42 | *GsNHL42* | *GlysoPI483463.13G263900* | 13 | 272 | 29.99915 | 10.01 |
| 43 | *GsNHL43* | *GlysoPI483463.13G298400* | 13 | 255 | 27.84143 | 10.24 |
| 44 | *GsNHL44* | *GlysoPI483463.14G036900* | 14 | 274 | 30.37919 | 9.65 |
| 45 | *GsNHL45* | *GlysoPI483463.14G166200* | 14 | 259 | 29.21918 | 10.15 |
| 46 | *GsNHL46* | *GlysoPI483463.15G014800* | 15 | 310 | 34.24409 | 9.95 |
| 47 | *GsNHL47* | *GlysoPI483463.16G088600* | 16 | 208 | 22.74147 | 9.23 |
| 48 | *GsNHL48* | *GlysoPI483463.16G179000* | 16 | 193 | 20.53805 | 9.30 |
| 49 | *GsNHL49* | *GlysoPI483463.18G044000* | 18 | 245 | 27.85761 | 9.97 |
| 50 | *GsNHL50* | *GlysoPI483463.18G046800* | 18 | 238 | 26.32375 | 8.06 |
| 51 | *GsNHL51* | *GlysoPI483463.19G161400* | 19 | 215 | 24.27526 | 9.74 |
| 52 | *GsNHL52* | *GlysoPI483463.19G161500* | 19 | 228 | 26.41171 | 9.34 |
| 53 | *GsNHL53* | *GlysoPI483463.19G161600* | 19 | 228 | 26.43265 | 9.34 |
| 54 | *GsNHL54* | *GlysoPI483463.19G161700* | 19 | 281 | 23.90965 | 9.81 |
| 55 | *GsNHL55* | *GlysoPI483463.19G161800* | 19 | 282 | 31.18464 | 8.22 |
| 56 | *GsNHL56* | *GlysoPI483463.19G209900* | 19 | 245 | 26.96660 | 9.77 |
| 57 | *GsNHL57* | *GlysoPI483463.19G216600* | 19 | 198 | 21.53948 | 9.89 |
| 58 | *GsNHL58* | *GlysoPI483463.20G108700* | 20 | 228 | 26.04367 | 9.36 |
| 59 | *GsNHL59* | *GlysoPI483463.20G179000* | 20 | 251 | 27.97813 | 9.74 |

elements, indicating the potential roles of *NHL* family genes in the MeJA and ABA signaling pathways. We also identified some response elements including MBS, LTR and TC-rich, which respond to drought, low temperatures and general stress, respectively. Collectively, these results strongly suggested that the roles of wild soybean *NHL* family genes are likely associated with plant abiotic stresses and hormone stimuli.

## Expression analysis of NHL genes in response to alkaline treatment in wild soybean

To assess the potential roles of *NHL* family genes participate in the defense responses towards alkaline stress, we generated a heat map of *NHL* family genes based on the wild soybean transcriptome data under alkaline stress. The results showed that 24 genes were differently induced under alkaline stress. Among them, 18 of *NHL* family genes were significantly up-regulated, while six genes showed down-regulation patterns (Fig. 4). To further confirm the expression of *NHL* family genes in response to alkaline treatment, we selected 12 of the up-regulated genes to detect their expression patterns under 50 mM NaHCO$_3$ stress by using qRT-PCR analysis. As shown in Fig. 5, the expression patterns of 11 up-regulated genes were roughly consistent with the transcriptome data under alkaline stress, except that *GsNHL29* had contrary results. In addition, the expression of *GsNHL9*, *GsNHL44*, *GsNHL45* and *GsNHL47* showed higher expression levels at 3 h point than the other genes (Figs. 5E, 5I, 5K). In conclusion, the qRT-PCR analysis confirmed the results that *GsNHL* family genes possibly participate in responses to alkaline stress.
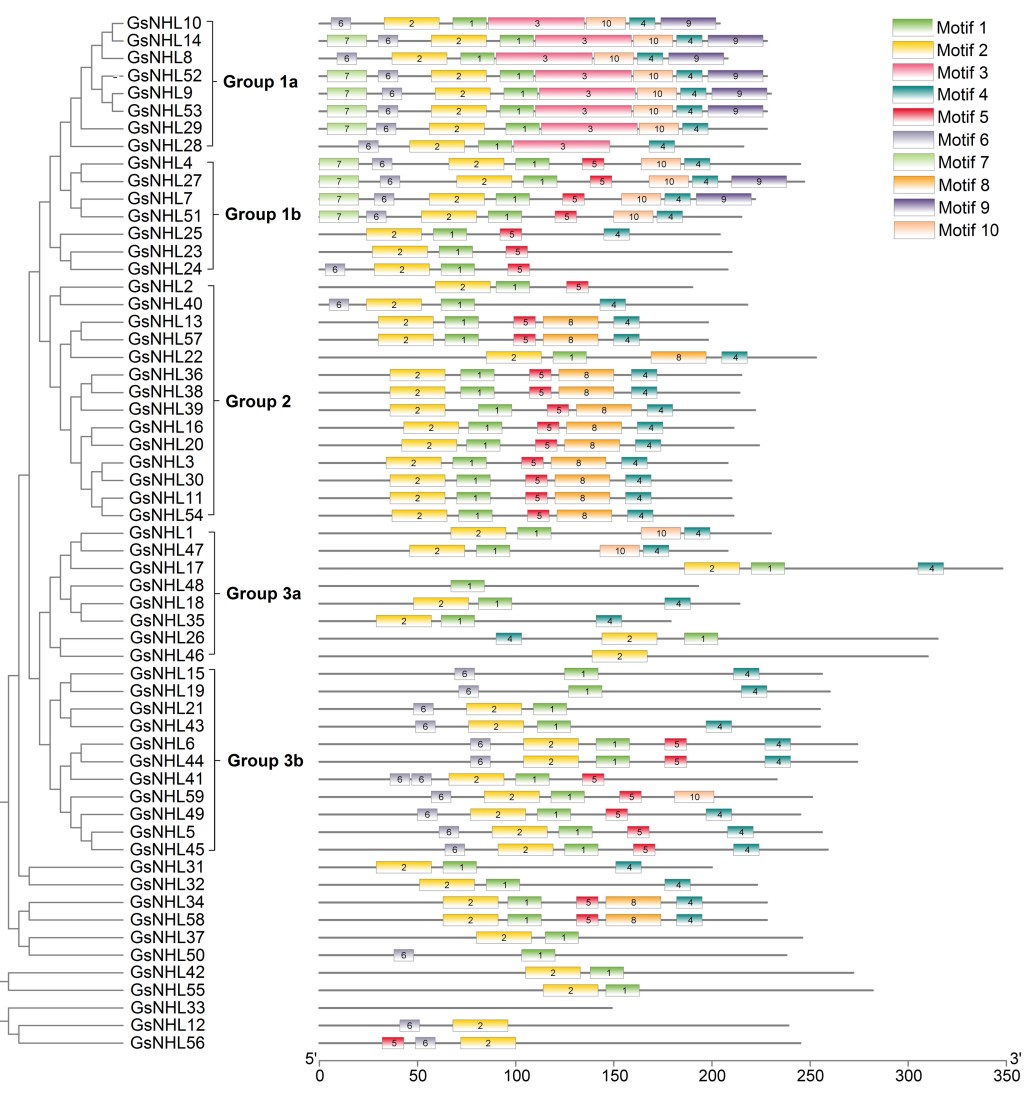

**Figure 1** **The phylogenetic and conserved domain analysis of *NHL* family proteins.** (A) The phylogenetic analysis of wild soybean *NHL* family proteins. The maximum-likelihood (ML) phylogenetic tree was constructed based on 1,000 replications for each branch. (B) The motif composition of *NHL* family proteins was identified using MEME online software, and the motif were displayed by boxes of different numbers and colors. The TBtools software was used to combine the conserved motifs and phylogenetic tree.

## Effects of different phytohormone treatments including ABA and MeJA on the expression of NHL genes

The plant hormones play regulatory roles in plant responses to various stresses. In this study, we found that the numbers of MeJA and ABA responsive elements in wild soybean *NHL* family genes were significantly larger than the other plant hormone responsive elements. Here, to explore if the *NHL* family genes could participate in ABA and MeJA signaling pathways, we analyzed the transcript expression levels of the 12 *NHL* family genes mentioned above under ABA and MeJA treatments by using qRT-PCR analysis. As shown

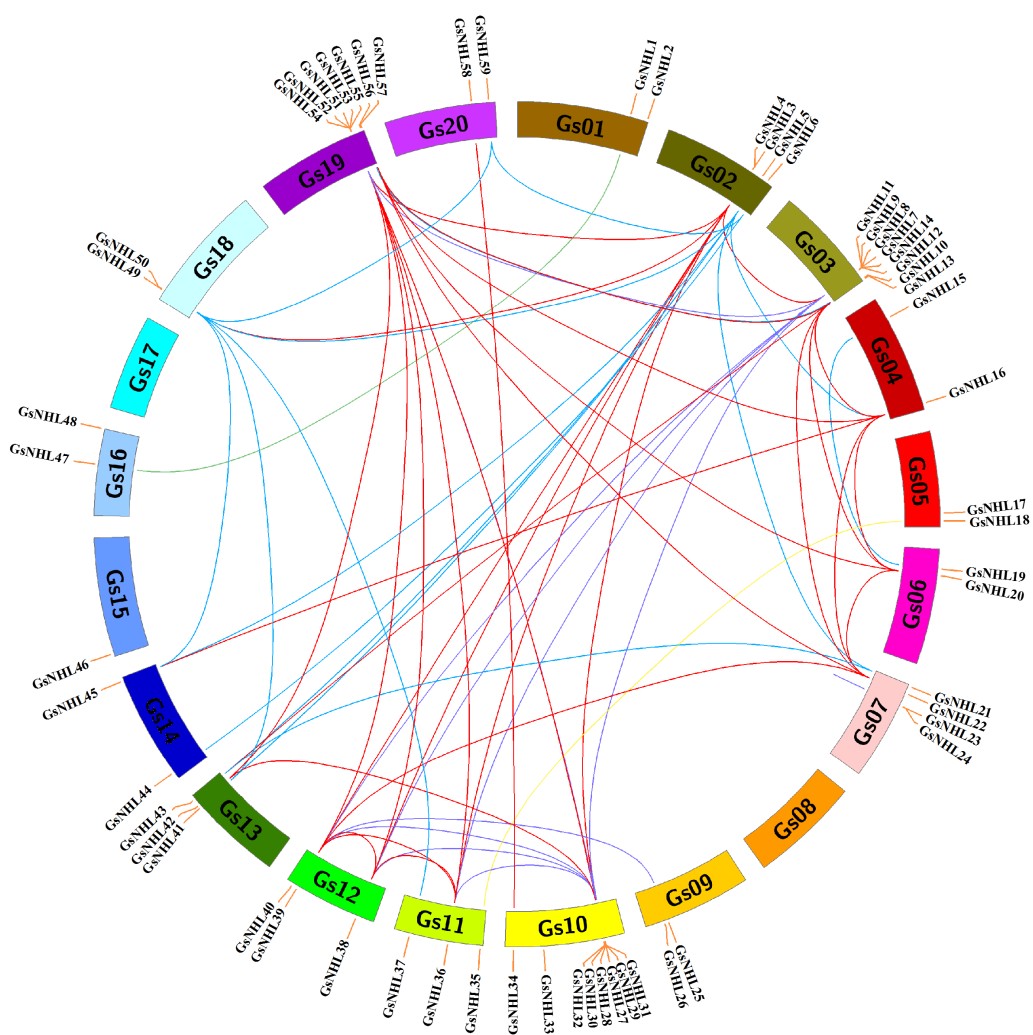

**Figure 2  Chromosomal locations and syntenic analysis of *NHL* family genes in wild soybean.** The Circos−0.69 software was used to generate the chromosomes as a circle. The pair of duplication genes were are identified and connected by different color lines.

in Fig. 6, nine genes were up-regulated under MeJA treatment and *GsNHL29* was down-regulated. In addition, *GsNHL6* and *GsNHL11* had contrary expression pattern at 1 h and 3 h (Figs. 6B, 6F). Under ABA treatment, only *GsNHL44* and *GsNHL51* were up-regulated and seven genes were down regulated (Fig. 7). *GsNHL4* had contrary expression pattern at 1 h and 3 h (Fig. 7A). *GsNHL6* and *GsNHL45* showed no significant expression changes (Figs. 7B, 7G). Collectively, these results indicated that *NHL* family genes participate in the ABA and MeJA signaling pathways and play different roles in response to the ABA or MeJA signaling pathway.

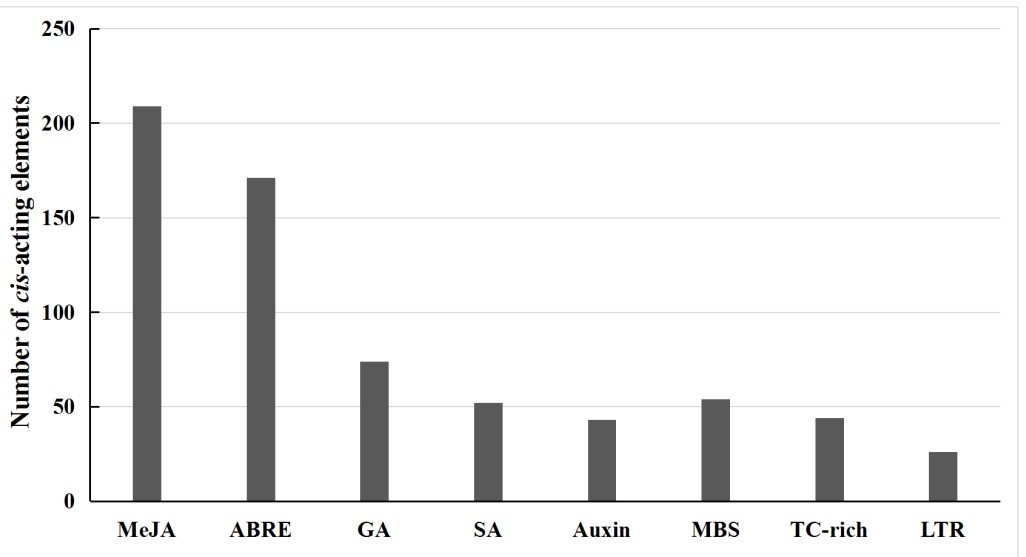

**Figure 3 Analysis of *cis*-acting elements of putative *NHL* family gene promoters related to hormones and abiotic stress responses.** The potential regulatory *cis*-acting elements were analyzed in the 3,000 bp upstream of translation start site by using the PlantCARE online software.

## DISCUSSION

Previously, studies have identified that *NHL* family genes are involved in plant development and pathogens attack resistance (*Bao et al., 2016*; *Chen et al., 2021*). Many *NHL* family genes have been identified in plant species, such as tomato, pepper and soybean (*Dormann et al., 2000*; *Liu et al., 2020b*; *Wang et al., 2022*). However, the wild soybean *NHL* family genes have not been identified, especially the roles of *NHL* family genes in regulating alkaline stress. Hence, this research was based on bioinformatics analysis about wild soybean *NHL* family genes in order to understand their structure and location, and mainly potential roles were investigated in response to plant hormones and alkaline stress treatments.

In this study, 59 wild soybean *NHL* family genes were identified in accordance with the soybean and Arabidopsis *NHL* related genes (Table 1). We found that NHL family proteins varied markedly in protein sequence length and molecular weight, indicating the divergent evolution in wild soybean *NHL* family genes. However, the high pI value showed NHL families are alkaline proteins.

Previous studies revealed that NHL protein family could be classified into six groups by investigating the relationship of soybean, Arabidopsis and rice(*Wang et al., 2022*). This result was consistent with our findings that wild soybean *NHL* family genes be divided into six groups (Fig. S1). On the basis of conserved motif analysis clustered, we found most of wild soybean *NHL* family genes could be classified into five groups, which was also consistent with the results of phylogenetic tree analysis of *NHL* family genes in soybean (*Wang et al., 2022*). In addition, each group almost shared a similar motif composition, which indicated that the groups may have similar roles in plant development progress (Fig. 1).

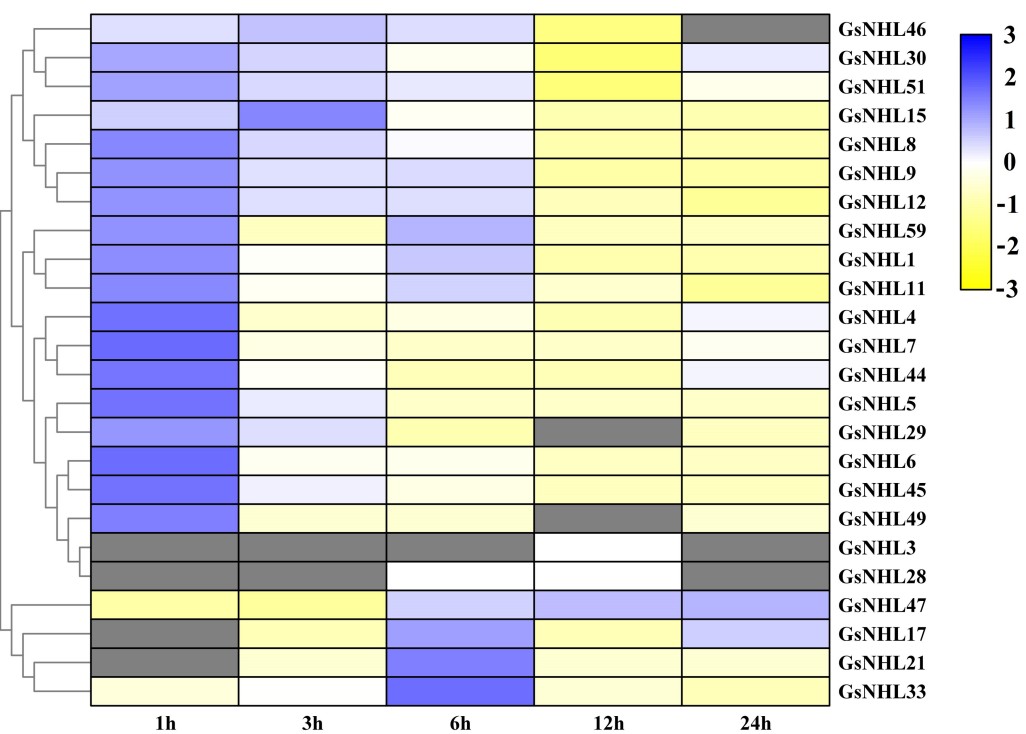

**Figure 4  Expression patterns of *NHL* family genes in response to alkaline stress.** The wild soybean transcriptome data were used to detect the expression pattern of NHL family genes under alkaline stress. The TM4: MeV4.9 software was used to generate the heat map. The blue and yellow colors represent high or low expression levels ($|$Log2 fold change$| > 2$, $P < 0.05$), respectively.

Further, the wild soybean *NHL* family genes from the same group were mostly located in different chromosomes and located near the edges of the chromosomes, suggesting a strategy to exert their functions in the whole wild soybean genome. The pairs of NHL syntenic paralogs also indicated that the wild soybean *NHL* family have been exhibited a high gene family expansion, which might play significant roles in gene functional diversity (Fig. 2).

In addition, we found that conserved motif 1, 3, 4, 5 and 10 belong to the LEA-2 domain (Fig. 1B, Fig. S2). This result also consistent with previous study (*Liu et al., 2020b*). Furthermore, we found that LEA-2 domain belongs to the LEA_2 subgroup which are widely known as a late embryogenesis abundant proteins and play significant roles under abiotic stress responses (*Jin et al., 2019*). For example, the rice LEA proteins showed accumulation during the salinity-triggered growth, while degradation in LEA proteins was observed during plant recovery from salt stress (*Chourey, Ramani & Apte, 2003*). The tea plant LEA genes were significantly induced under stress conditions, such as drought, ABA, low and high temperature (*Jin et al., 2019*). Overexpression of *IpLEA* could show high tolerance to salt and drought stress in *Ipomoea pescaprae* by mediating water homeostasis and as a reactive oxygen species scavenger (*Zheng et al., 2019*). Thus, this evidence indicated the potential roles of wild soybean *NHL* family genes in response to environmental stresses.

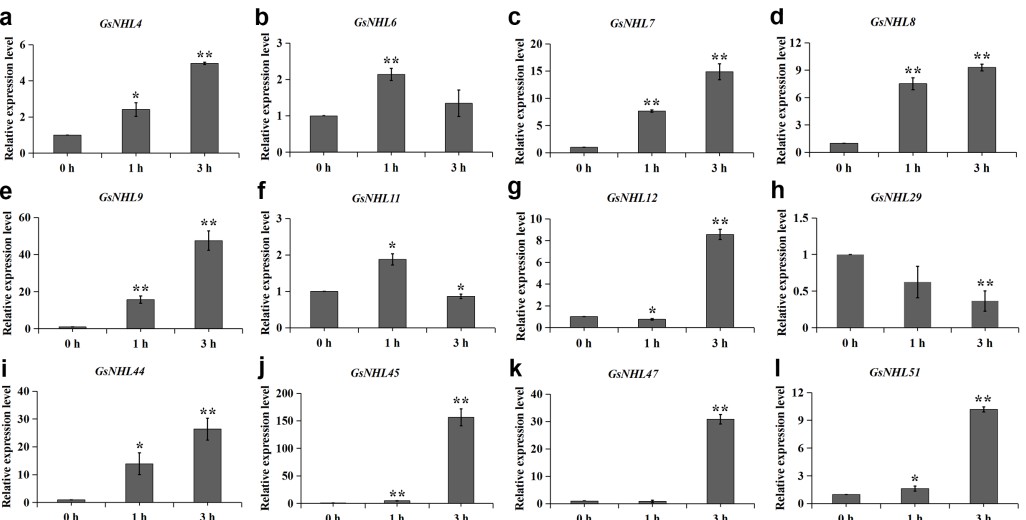

**Figure 5 Expression analysis of wild soybean *NHL* family genes in response to alkaline stress.** (A–L) The wild soybean seedlings were treated with 50 mM NaHCO$_3$ for 0, 1 and 3 h. The qRT-PCR results were analyzed using the $2^{-\Delta\Delta CT}$ method and Student's *t*-test.

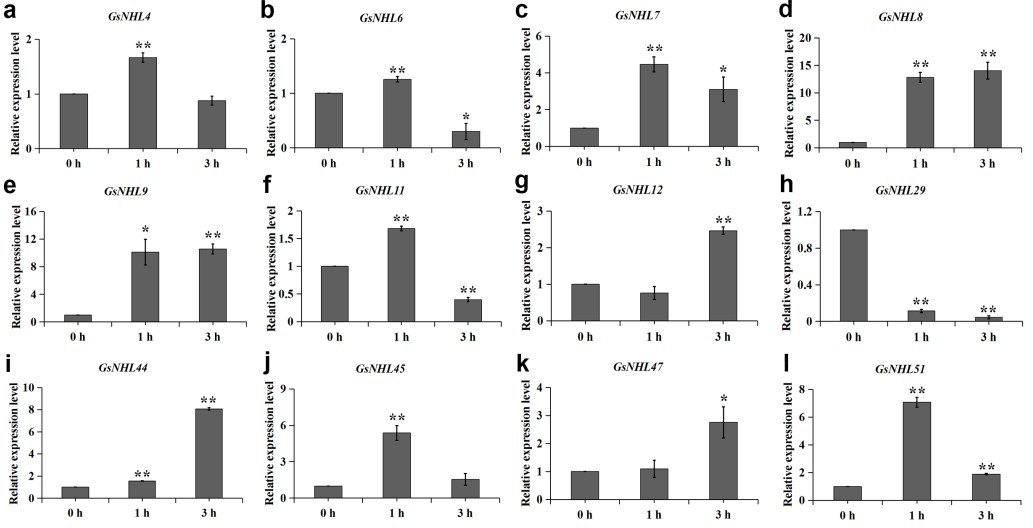

**Figure 6 Expression analysis of wild soybean *NHL* family genes in response to MeJA.** (A–L) The wild soybean seedlings were treated with 50 μM MeJA for 0, 1 and 3 h. The qRT-PCR results were analyzed using the $2^{-\Delta\Delta CT}$ method and Student's *t*-test.

The *cis*-acting regulatory elements play important roles as molecular switches to control various biological processes, including hormonal and various stress responses (*Sun et al., 2021*). The *cis*-acting regulatory elements analysis showed that the promoter regions of wild soybean *NHL* family genes contain a variety of putative hormone-related and abiotic stress responsive elements (Fig. 3). Previous studies have been shown that *NHL* family genes participate in the plant hormone-mediated pathways. For example, overexpression

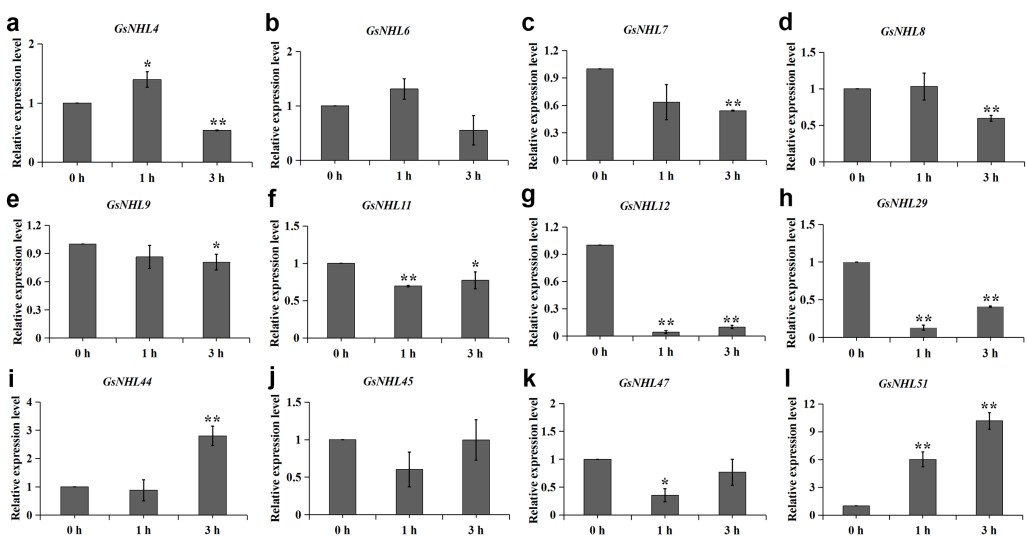

**Figure 7** **Expression analysis of wild soybean *NHL* family genes in response to ABA.** (A–L) The wild soybean seedlings were treated with 50 µM ABA for 0, 1 and 3 h. The qRT-PCR results were analyzed using the $2^{-\triangle\triangle CT}$ method and Student's $t$-test.

of *AtNHL1* and *AtNHL8* in soybean could enhance plants resistance to *Heterodera glycines* by mediating the jasmonic acid and ethylene pathways, confirming the roles of these genes in plant defense response (*Maldonado et al., 2014*). *NHL6* participates in the abiotic stresses-induced ABA signaling at seed germination and early seedling stages in Arabidopsis (*Bao et al., 2016*). *BnNHL18A* could be significantly induced by NaCl, ethephon, methyl jasmonate or salicylic acid treatment in *Brassica napus* (*Lee et al., 2006*). Also, LTR is a *cis*-element responsive to low-temperature stress (*Brown et al., 2001*). TC-rich *cis*-element has been identified to be involved in stress mediated plant defense responses (*Sazegari, Niazi & Ahmadi, 2015*). In conclusion, these evidence strongly suggested that *NHL* family genes may be involved in stress resistance and plant hormones responses in wild soybean.

Alkaline stress is one of the most harmful abiotic stresses, which leading to a series of regulatory mechanisms in plants, such as ion balance, osmotic adjustment, pH regulation, and ROS scavenging mechanisms. Previously, we identified a highly adaptable saline-alkali soil tolerant wild soybean line which can survive well in the saline-alkali soil. Then, we explored the differentially expressed genes of wild soybean seedlings treated with 50 mM $NaHCO_3$ by RNA sequencing (*DuanMu et al., 2015*). In this study, we mainly intend to explore the potential roles of wild soybean *NHL* family genes in response to alkaline stress. According to the transcriptome data, a total of 24 genes were significantly induced under alkaline stress (Fig. 4), and qRT-PCR confirmed the results that wild soybean *NHL* family genes may play positive role in response to alkaline stress (Fig. 5).

During abiotic stress responsive processes, plant hormones such as MeJA, ABA, SA, GA and Auxin also play important roles and have cross talks in signal transduction pathways (*Ku et al., 2018*). In our previous study, we also identified that plant hormones have crosstalk with plant alkaline stress resistance response. For example, the wild soybean

gene *ERF71* could regulate endogenous auxin accumulation when plants treated with alkaline solution (*Yu et al., 2017*). The *TIFY10* gene could act as a regulator in response to alkaline stress and jasmonate signaling in wild soybean (*Zhu et al., 2011*). On the other hand, we found that wild soybean *NHL* family genes comprised of a variety of putative hormone-related responsive elements, and the numbers of MeJA and ABA responsive elements were significantly larger than others. Thus, we concluded that the wild soybean *NHL* family genes have crosstalk with MeJA or ABA signal transduction under alkaline stress. qRT-PCR analysis showed that nine genes were up-regulated under MeJA treatment, and these genes were all up-regulated under alkaline stress (Fig. 6). This finding is consistent with the previous studies that wild soybean *TIFY10a* overexpression lines enhanced the alkaline stress resistance and also increased the jasmonate content of the transgenic alfalfa (*Zhu et al., 2014*). However, in comparison with MeJA treatment, qRT-PCR analysis showed a different expression, in which only two wild soybean *NHL* family genes were up-regulated and seven genes were down-regulated under ABA treatment (Fig. 7). For example, *GsNHL7*, *GsNHL8*, *GsNHL9*, *GsNHL12* and *GsNHL7* were up-regulated under alkaline stress and MeJA treatment, while were down-regulated under ABA treatment. In addition, our previous studies identified that overexpression of wild soybean *NAC019* or *SKP21* in *Arabidopsis* could contribute to alkaline stress tolerance, but reduced ABA sensitivity(*Cao et al., 2017*; *Liu et al., 2015*). In conclusion, these results speculated that wild soybean *NHL* family genes have crosstalk with MeJA or ABA signal transduction under alkaline stress, and some genes may display different roles in ABA or MeJA signal transduction in response to alkaline stress.

## CONCLUSIONS

In conclusion, in this study, we identified 59 potential *NHL* family genes in wild soybean. We identified the phylogenetic relationship, conserved domains, gene duplication events and *cis*-acting elements in promoter regions. We also confirmed that wild soybean *NHL* family genes may play important regulatory roles in response to alkaline stress, ABA and MEJA treatment. Taken together, our results established a foundation for characterization of wild soybean *NHL* family genes in response to alkaline stress, ABA and MEJA treatment. However, the function analysis of up-regulated genes under ABA, MEJA or alkaline stress, such as *GsNHL4*, *GsNHL44* and *GsNHL51*, is of great significance in the future. On the other hand, more work is required for exploring the potential roles of NHL family genes, especially the roles in crosstalk with MeJA or ABA signal transduction pathways under alkaline stress in wild soybean.

### Abbreviations

| | |
|---|---|
| **NHL** | NDR1/HIN1-like |
| **qRT-PCR** | quantitative real-time PCR |
| **MW** | molecular weight |
| **pI** | isoelectric point |
| **ABA** | abscisic acid |
| **MeJA** | methyl jasmonate |

| SA | salicylie acid |
| GA | gibberellin |
| MBS | drought |
| LTR | low temperature responsive |
| TC-rich | defense and stress responsive |

### Funding

This work was supported by the Natural and Science Foundation of China (grant number 32001454), the Natural Science Foundation of Heilongjiang Province (grant number LH2020C068), the China Postdoctoral Science Foundation (2022M711431), the Scientific Research Project of Innovative Talents in Heilongjiang Province (grant number UNPYSCT-2020125) and the Joint guidance project of Department of Science and Technology of Heilongjiang Province (LH2021C052). The funders had no role in study design, data collection and analysis, decision to publish, or preparation of the manuscript.

### Grant Disclosures

The following grant information was disclosed by the authors:
Natural and Science Foundation of China: 32001454.
Natural Science Foundation of Heilongjiang Province: LH2020C068.
China Postdoctoral Science Foundation: 2022M711431.
Scientific Research Project of Innovative Talents in Heilongjiang Province: UNPYSCT-2020125.
Joint guidance project of Department of Science and Technology of Heilongjiang Province: LH2021C052.

### Competing Interests

The authors declare there are no competing interests.

### Author Contributions

- Xu Zhang performed the experiments, prepared figures and/or tables, and approved the final draft.
- Yongguo Xue conceived and designed the experiments, prepared figures and/or tables, and approved the final draft.
- Haihang Wang performed the experiments, prepared figures and/or tables, and approved the final draft.
- Zaib_un Nisa analyzed the data, authored or reviewed drafts of the article, and approved the final draft.
- Xiaoxia Jin performed the experiments, authored or reviewed drafts of the article, and approved the final draft.
- Lijie Yu conceived and designed the experiments, authored or reviewed drafts of the article, contributed reagents, and approved the final draft.

- Xinlei Liu conceived and designed the experiments, authored or reviewed drafts of the article, contributed reagents, and approved the final draft.
- Yang Yu performed the experiments, prepared figures and/or tables, and approved the final draft.
- Chao Chen conceived and designed the experiments, analyzed the data, prepared figures and/or tables, authored or reviewed drafts of the article, and approved the final draft.

## Data Availability

The raw data are available in the Supplementary Files.

## Supplemental Information

Supplemental information for this article can be found online at http://dx.doi.org/10.7717/peerj.14451#supplemental-information.

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
