# Peer review of "Genome-wide identification and characterization of NHL gene family in response to alkaline stress, ABA and MEJA treatments in wild soybean (Glycine soja)"

_PeerJ, doi:10.7717/peerj.14451_

## Round 0.1 · original submission · Major Revisions

The article is not accepted in its present form. Substantial changes are required. Kindly incorporate all the changes as suggested by the reviewers.

Reviewer 1 ·

Basic reporting

The work of Zhang et al. presents the initial search and characterization of the NHL gene family in Glycine soja. The novelty of the work lies in the species studied. The methodology, approaches, and way of presenting the results mimic previous work on NHL search and characterization in other plant species (e.g. https://doi.org/10.1038/s41438-020-0318-0).

Experimental design

- You may need to add more recent reports for the NHL gene family in the introduction.
- “To identify all potential genes encoding NHL family genes in the wild soybean genome, a
Hidden Markov model was first established by using the amino acid sequences of known
Arabidopsis NHL family genes as queries”. To make a comprehensive search, you may need to generate Hidden Markov model using NHL amino acid sequences from multiple plant species.
- Why did you use Mega 5.0? you may need to use advanced versions of this software.
- Why did you use the neighbor-joining method and why not other methods?
- You may need to discuss the importance of sequence lengths, molecular weights (MW), and theoretical isoelectric points (pI).
- You performed phylogenetic analysis using NHL protein sequences of only wild soybean. You may need to include multiple plant species in phylogenetic analysis to infer representative evolutionary relationships.
- You may need to give a reason for the selection of the 3 Kb promoter sequence for analysis.
- - Since there is a report of NHL genes in a nearby species (https://www.nature.com/articles/s41438-020-0318-0#Abs1), it would be interesting to see if the NHLs they found here are homologous to those reported by Liu et al., 2020. Also, it would be interesting to see the synteny of these genes in related species.
- You may need to provide more details of Transcriptome data.
- There were only 6 motifs in protein sequences or did you just select 6 motifs?

Validity of the findings

After re-analyzing the data, as mentioned above, Please correct the "results" and "discussion" sections accordingly.

Additional comments

Few sentences need correction like
- A total of 65 pairs of wild soybean NHL syntenic paralogs were identified, indicating this family existed a high gene family expansion (Fig. 2).

·

Basic reporting

This manuscript reports an identification and characterization of NHL family genes of Glycine soja, through the description of regulatory regions and shows a relationship of these genes with abiotic stress response, particularly under alkaline conditions, which is relevant and contributes to fill the gaps in knowledge about plant mechanisms under stress stimuli. The authors present a good article structure, with good quality and relevant figures and tables, and provide enough raw data. I suggest paying attention to the following points that should be improved.

The English language needs to be improved to ensure readers clearly understand the text and a professional English. Some examples where the language could be improved include lines 57, 99, 145,– the current phrasing makes comprehension difficult.

The introductory section did not reflect a robust review of the literature and should be improved by including more contextualization and using more relevant studies. In particular, I found it necessary to delve into NHL gene family using studies with soybean, such as, “NDR1/HIN1-like genes may regulate Glycine max seed germination under chilling stress through the ABA pathway” (Wang et al. 2022) https://link.springer.com/article/10.1007/s10725-022-00894-x
Likewise, it is important to say why the authors chose glycine soja as an object of study.
I recommend to looking for more studies on NHL and alkaline stress, and be careful about the claim “However, the roles of NHL family have not explored under alkaline stress”.
For example, Liu et al. 2020 reported NHL genes presented differential expression under salt and alkaline stress in “Genomics-assisted prediction of salt and alkali tolerances and functional marker development in apple rootstocks” https://bmcgenomics.biomedcentral.com/articles/10.1186/s12864-020-06961-9

I suggest the authors include comparisons with other soybean reports in the discussion section, as I mentioned before, and it is important to check if there are others similar studies.
On the other hand, in discussion section, the authors mentioned they found LEA domains in soybean NHLs, which is reported earlier in other studies. I suggest to include this information in the introduction, and reporting that the author´s findings are consistent with prior knowledge.

Experimental design

The manuscript described an experimental design and appropriate methods to achieve the objectives. However, I recommend using a new version of MEGA and more appropriate methods such as Maximum likelihood or Bayesian to analyze the phylogenetic relationship. Further details on the transcriptomic data are lacking. I suggest including a brief description of the experiment to produce those libraries, and it is very important to detail how those data were analyzed, so that they are reproducible.

Validity of the findings

Considering that functional genomics analysis will validate the results, I encourage the authors to mention their perspectives and what is recommended for future studies as next steps, in the conclusions section.

Additional comments

Other comments and suggestions are presented throughout the text of the PDF.

---

## Round 0.2 · accepted · Accept

All the comments have been incorporated. The manuscript is now suitable for publication. Please see the annotated PDF from Reviewer 2 for some minor edits.

Reviewer 1 ·

Basic reporting

NA

Experimental design

NA

Validity of the findings

NA

Additional comments

The authors have significantly improved the manuscript and have responded to all my comments. I think the manuscript is ready for publication.

·

Basic reporting

no comment

Experimental design

no comment

Validity of the findings

no comment

Additional comments

I think the manuscript has been improved, it has become clearer and the information has been completed. Minor commets were left on the PDF manucript.